
# Data Assimilation of Argos profiles in North-west Pacific Model

**Zhaoyi Wang[1,2,3], Andrea Storto[2], Nadia Pinardi[2], Guimei Liu*[1,3], HuiWang[1,3]**

[1]National Marine Environmental Forecasting Center of China (NMEFC), Beijing, China (100081)

[2]Centro Euro-Mediterraneo per i Cambiamenti Climatici (CMCC), Bologna, Italy (40139)

[3]Key laboratory of Research on Marine Hazards Forecasting, National Marine Environmental Forecasting Center, Beijing, 100081, China

*Corresponding author, E-mail: liugm@nmefc.gov.cn

_______________________________________________________________________

***Abstract:***

Based on a novel specification of the background error covariance applying to Argos profiles assimilation, an oceanographic three-dimensional variational (3DVAR) data assimilation scheme is set up in the Regional Oceanic Model system (ROMs). Temperature and salinity data extracted from Argos profiles in 2006 have been assimilated into the North-West Pacific Model (NWPM). The quality control is done by comparing background estimation with observations in 2006.Firstly, the assimilated results are compared with merged *in-situ* data, Sea Surface Temperature (SST) derived from satellite data and reanalysis salinity data. It is found that assimilation of Argos profiles can improve the model results of SST and salinity. Secondly, the Root Mean Square (RMS) difference between model and Argos profiles is analyzed. For the tropic Pacific, the range of RMS temperature (salinity) error are less than 0.83 ℃ (0.11 PSU), decreasing ~23.2% (~18.8%) by comparing with the experiments without data assimilation. For the sub-tropic Pacific Ocean, the RMS of temperature (salinity) is less than 1.43 ℃ (0.135 PSU) and it also shows a decreasing trend after assimilation. It's indicated that the 3DVAR method works well in ROMs and can be used for the operational forecasting systems.

***Keywords:*** *Data Assimilation, Argos, North-west Pacific*

----------------------------------------------------------------------------------------------------




## 1. Introduction


The Chinese Global operational Oceanography Forecasting System (CGOFS) running
at National Marine Environmental Forecasting Center of China (NMEFC) is used to
predict properties of global ocean, such as temperature, salinity, current, wave and sea
ice. The operational North-West Pacific Model (NWPM) is a regional model of
CGOFS consisting of a suite of nested model configurations, which produce daily
analysis and forecast, out to 5 days ahead, of the ocean variables, and provide the
nested configuration for East China Sea Model (ECSM) and South China Sea Model
(SCSM). The model component of NWPM is based on the Regional Ocean Model
System (ROMS) which is a free-surface, primitive equation ocean circulation model
formulated using terrain-following coordinates.
Model forecast requires the specification of initial conditions, and the accuracy of the
forecast depends on the accuracy of the initial conditions. Data assimilation is a
widely used and proved effective way to produce best estimates of the state of the
physical system by integrating observations into prognostic model. Over the past few
decades, many data assimilation methods have been developed for combining model
and observational data. These can broadly split into three approaches: Kalman Filter,
generally known as sequential schemes (Daley, 1991); Optimal Interpolation; and
variational methods (Lorenc, 1986), which are based on minimization of a cost
function that measures the differences between the model and the observations. The
Ensemble Kalman Filter (EnKF) was introduced by Evensen *et al.* (2003). Because of
the computational requirements limitation, the EnKF is not suitable for operational
forecasting system. As an approximation of EnKF, Ensemble Optimal Interpolation
(EnOI) scheme has been applied to ROMS to assimilate the along track Sea Level
Anomaly (TSLA) (Lv *et al.*, 2013). ROMS also is equipped with the four-dimensional
variational assimilation (4DVAR) method (Tshimanga *et al*, 2008; Moore *et al.*, 2011a,
2011b), which isn't used in the operational forecasting system considering the
computational requirements. With considering the 3DVAR is the soundest path to the
ultimate development of more advanced data assimilation systems, Li *et al* (2008) has


developed a 3DVAR approach for ROMS independently.
Three-dimensional variational (3DVAR) data assimilation method is a widely used
method in oceanic operational forecasting systems (e.g. Li et al, 2008). In this study
we applied an oceanographic three-dimensional variational data assimilation scheme
called OCEANVAR (Dobricic and Pinardi, 2008) to ROMS to assimilate the
Temperature and Salinity profiles from Argos. In order to illustrate and evaluate the
performance of the assimilation scheme, it was applied to the north-west pacific with
an eddy-resolving resolution. This system will be used in the future to augment the
quality of initial conditions for daily forecasts that has started to produce on
CGOFSv1.0.
The paper is organized as follows. The section 2 describes the components of the data
assimilation scheme for assimilating Argos profiles in the North-west Pacific. The
results from data assimilation are presented in the section 3, with focusing on the
performance of 3DVAR and multivariate properties. Finally, section 4 presents
conclusions.
**2. Model and data**
**2.1 model configuration**
The ROMS (Shchepetkin and McWilliams 2005; Malcolm et al., 2009) used in this
paper is a free-surface and primitive equation ocean circulation model formulated
using terrain-following coordinates, which is widely used in oceanic studies (Wang, et
al. 2012, Lv et al,.2014) . The model domain in this study is North-west Pacific Ocean
that extends from 8°S to 52°N and from 99°E to 160°E, as shown in Figure 2.1. The
horizontal resolution is 1/20° in both zonal and meridional directions with a total
horizontal grid points of 1098×1084 . In vertical, there are 30σ layers. The maximum
depth is set to 7000 m to keep the pattern stable. The bathymetry used here is derived
from GEBCO (General Bathymetric Chart of the Oceans), a global 30 arc-second
gridded bathymetry, which was supplied by the Intergovernmental Oceanographic
Commission and International Hydrographic Organization. To reduce the influence of
the seamount on model stability, the bathymetry is smoothed appropriately.
Considering the effects of an open boundary on simulation, southern, western and



northern boundaries are set as open boundaries, of which water level and velocity are
also obtained from SODA. The internal model time step is 300s and the external
model time step is 10s. The Yangtze River, Pearl River and Mekong River use
monthly mean runoff values in the model.
The model is spined-up for 10 years with the COADS (Comprehensive
Ocean-Atmosphere Data Set) monthly climatological mean air-sea flux to get an
initial state. From this initial condition, the model is forced by the NCEP/NCAR
Reanalysis2 4× daily data to simulate condition for the period of 1990-2005. The
initial conditions for both the control and the assimilation runs are provided by the
simulated ocean state at the end of 2005. In additional, the control test for 2006
without data assimilation provides a basis for comparison.

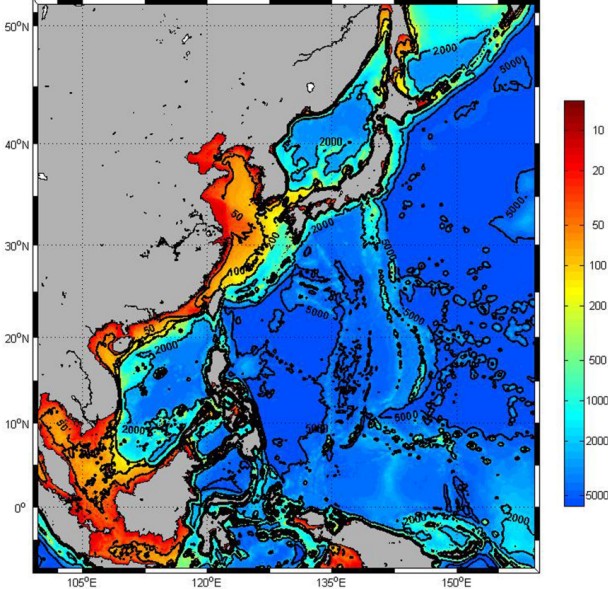


*Fig.2.1. Bathymetry of the North-west Pacific in the numerical model (depths in*
*meters)*
**2.2. Assimilation algorithms**
In recent years, progress in ocean data assimilation has been enabled along with the
advances in computing machinery and mathematical roots. The basic goal of the
ROMS 3DVAR system is to provide an "optimal" estimate of the true oceanic state at
analysis time through solving the assimilation problem by minimizing the prescribed





cost function (Ide et al. 1997)

$$\mathcal{J} = \mathcal{J}_b + \mathcal{J}_o$$

$$= \frac{1}{2}(x - x^b)^T \mathbf{B}^{-1}(x - x^b) + \frac{1}{2}[H(x) - y^o]^T \mathbf{R}^{-1}[H(x) - y^o] \qquad (2.1)$$

Where $x$ is the unknown ocean state, equal to the analysis $x^a$ at the minimum of $\mathcal{J}$;
$x^b$ is the background, which is ana priori estimate of the state of the ocean; $y^o$ is the
vector of the observations; $y = H(x)$ is transform the gridded analysis $x$ to
observation space, with $H$ is the linear observation operator; and $B$ and $R$ are the
covariance matrices of the backgroundand observational errors, respectively. The Eq.
(2.1) is linearized around the background state (e.g. Lorenc, 1997) into the following
form:
$$\mathcal{J} = \frac{1}{2}\delta x^T B^{-1} \delta x + \frac{1}{2}(H\delta x - d)^T R^{-1}(H\delta x - d) \qquad (2.2)$$

Where $d = [y^o - H(x_b)]$ is the misfit, $H$ is the linearized observations operator
evaluated at $x = x_b$ and $\delta x = x - x_b$ are the increments. The minimization problem
is defined on the field of increments in Eq. (2.2) which has a single minimum.
The 3DVAR system uses vertical Empirical Orthogonal Functions (v-EOFs) to
represent vertical modes of the background-error correlation matrix. The new v-EOFs
are considered time with monthly timescales.
The use of adjoint operations, which can be regarded as a multidimensional
application of the chain-rule for partial differentiation, permits efficient calculation of
the gradient of the cost function. The Quasi-Newton L-BFGS (Limited-memory
Broyden-Fletcher-Goldfarb-Shanno, Byrd et al., 1995) is used to efficiently combine
cost function, gradient and the analysis information to produce the "Optimal"
analysis.
The data assimilation systems, such as OI, EnKF, and 3DVAR, have led to improved
forecast scores relatively quickly. The practical advantages of VAR system over other
methods are listed below. Firstly, the VAR solution uses all observations
simultaneously, compared to the OI technique for which the process of data selection
into artificial sub-domains is required; secondly, asynoptic data, such as satellite and
radar observations, can be assimilated near its validity time; Thirdly, balance, for





weak geostrophy and hydrostatic, constraints can be built into the preconditioning of
the coast function minimization. Even with such practical advantages, VAR system
still shows some weaknesses in real practice. Firstly, given both imperfect
observations and prior (e.g. background) information as inputs to the assimilation
system, the quality of the output analysis depends crucially on the accuracy of
prescribed errors. Secondly, although the variational method allows for the inclusion
of linearized dynamical/physical processes, in reality, real errors in the prediction
system may be highly nonlinear, which limits the usefulness of variational data
assimilation in highly nonlinear regimes, e.g. the convective scale or in the tropics.

**3. Model Validation**

In this section, we show the results of the assimilation of in situ data from January
2006 to March 2007. We discuss the validation of the analysis to evaluated the
performance of the assimilated model, wherein the simulated fields and the analysis
fields are called SF and AF, respectively.

**3.1. Consistency**

Consistency checks were carried out by comparing the AF monthly mean SST with
the Merged satellite and in situ data Global Daily Sea Surface Temperatures
(MGDSST).
Fig. 3.1 shows monthly mean average SST in January, April, July and October of
2006 from simulation and MGDSST respectively. The model SST (Fig. 3.1(b)) is
consistent with that derived from MGDSST (Fig. 3.1(a)). In subtropical basins,
temperature is generally high near the western boundary. While in sub-polar basins,
the zonal temperature gradient reverses sign, with low temperature in the western
basin. In addition, SST is reduced in pole-ward direction, with high temperature in the
equatorial Pacific and low temperature in the polar Pacific. Simulated SST is higher in
summer and lower in winter, compared to MGDSST (Fig. 3.1(d)). SST is generally
similar to the MGDSST in subtropical basins, meanwhile shows the pattern of high in
summer and low in winter, to the north of 40°N. The ocean model at high spatial
resolution can reasonably simulate the distribution of the warm pool.
In order to understand the effect of data assimilation for temperature, Fig. 3.1(b) and



Fig. 3.1(c) show the differences between the observation and analysis or simulation,
respectively. First of all, the AF SST errors are generally smaller than the
corresponding SF, thus closer to the MGDSST observations. The SST has substantial
improvement in the South China Sea, the East China Sea and the Subtropics Pacific
after data assimilation. The SST error in Kuroshio Extension also has improvement in
the AF in agreement with the in-situ and satellite observations.
Fig. 3.2 shows the salinity profiles of simulations and observations, which are derived
from the EN4.0.2 dataset at 150 m (1°×1°, Good *et al.*, 2013). Although the AF
salinity for the tropic Pacific is less saline than the observation, the AF salinity for the
sub-tropic Pacific is very similar with the observation, which means that AF salinity
can catch main characteristics of actual salinity patterns.
Temperature section of 136°E is presented in Fig. 3.3, which shows some significant
qualitative differences. Fig. 3.3(b) shows the section with the AF and Fig 3.3(a) shows
the corresponding observation of EN4.0.2. Fig. 3.4(a) and Fig. 3.4(b) show the
salinity section with AF and EN4, respectively. The assimilation is capable of
modifying the vertical extension of Pacific.



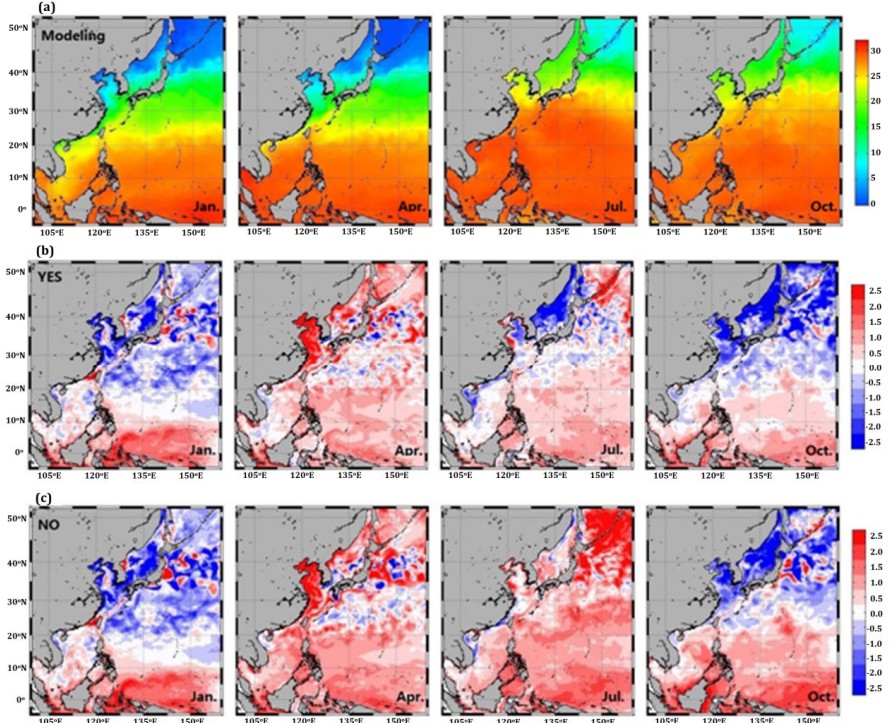


*Fig. 3.1. Monthly mean temperature at 150 m depth (℃) for January, April, July and October*
*2006.(a)The SST of AF; (b)difference between MGDSST and AF; (c) difference between*
*MGDSST and SF.*

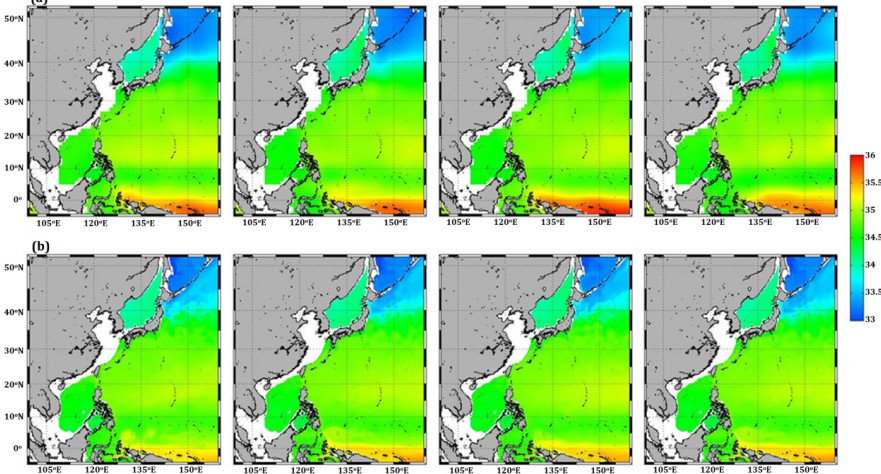


*Fig. 3.2. Monthly mean salinity at 150 m depth (in psu) in January, April, July and October*
*2006.(a) EN4, (b) AF*


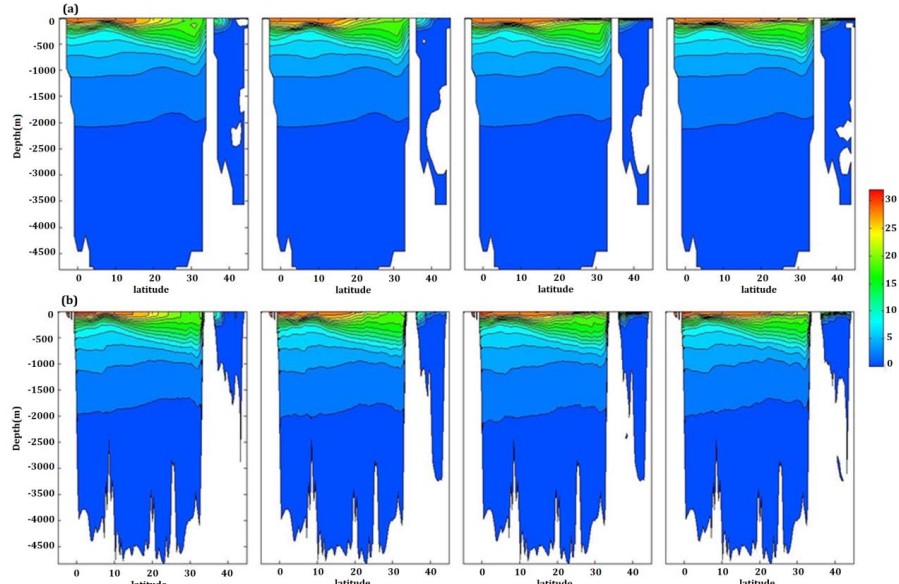


*Fig. 3.3. Temperature (in ℃) section of 136°E for(a) EN4 and (b) AF*

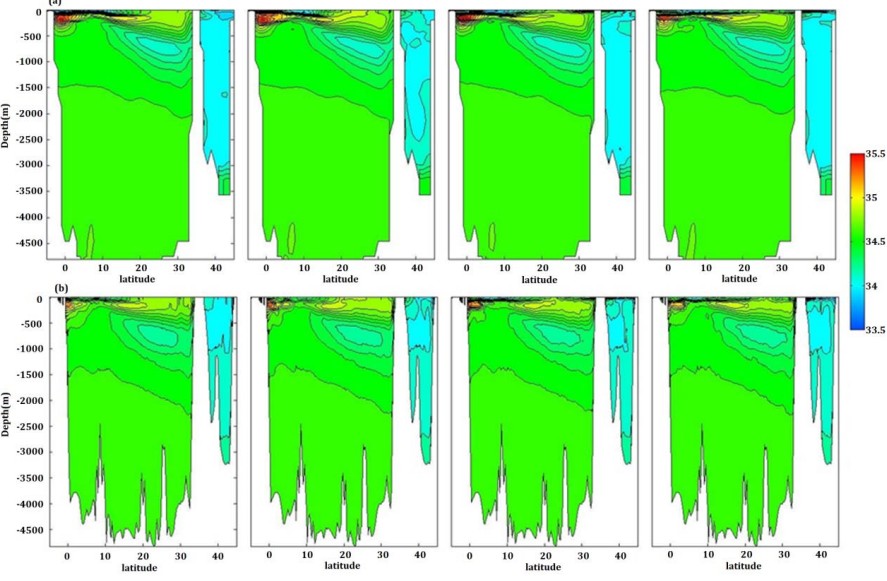

*Fig. 3.4. Salinity (in psu) section of 136°E for(a) EN4 And (b) AF*

Finally, a qualitative analysis of the impact of the assimilation in the modeling region
is shown as follow. In Fig. 3.5(a), (b), (c), and (d), the differences in the study region
of before and after assimilation profiles are shown together with the observed profiles.
As shown, the profiles after assimilation are between the SF profiles and the



observations. The comparison results show that the data assimilation system is
capable of correcting the model, with an effect of bringing the model closer to the
observations.

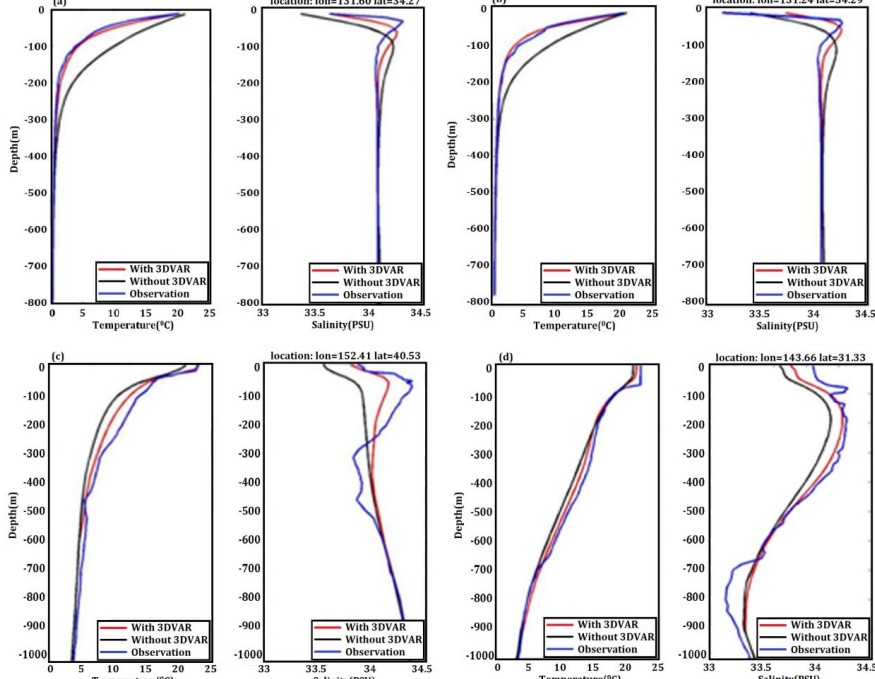


*Fig. 3.5. The vertical profiles for temperature (in ℃) and salinity (in psu), where, red line*
*stands for AF, black line stands for SF and blue line stands for observation; vertical and*
*horizontal axes are depth (m) and temperature or salinity, respectively.*
**3.2 Accuracy**
Some quantitative information on the analysis quality can be obtained by comparing
the analyses with the observations at the observation locations. We use the misfit error
which is the difference between the observation and the SF or AF to analyze the
improvement of the model solution due to the regular data assimilation, although the
data isn't independent. The Root Mean Square (RMS) between the SF or AF values
and the observation values is defined as:
$$RMS = \sqrt{\frac{1}{n}\sum_{i=1}^{n}(\varphi_m - \varphi_o)^2} \qquad (3.1)$$

where, $\varphi_m$ and $\varphi_o$ stand for model and observation values of temperature or
salinity respectively, n is the number of observation during the assimilation cycle.





Fig. 3.6 shows the RMS of temperature and salinity misfits which calculated as Argo
observation minus background value. The experiment with 3DVAR analyses has a
lower RMS of misfits than the experiment without assimilation. Furthermore, the
RMS with the assimilation becomes practically insignificant in deeper layer of the
ocean. As shown in the left panel of Fig. 3.6, the RMS of temperature misfits has the
maximum at ~100 m depth which approximately corresponds to the depth of the
mixed layer. The RMS of temperature misfits is relatively small close to the surface,
probably due to the fact that surface temperature is relaxed towards MGDSST
observations in both experiments. As shown in the right panel of Fig. 3.6, the RMS of
salinity misfits is significantly reduced after assimilation, especially at the depths
between ~200 m and ~400 m. However, the RMS of misfits increases towards the
surface in both experiments. The reason can be explained by the surface water and salt
flux, which is computed by relaxing the surface salinity towards climatology in the
model.

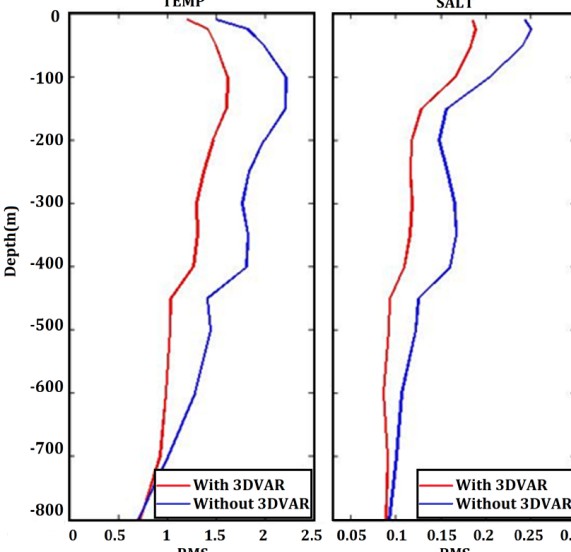


*Fig. 3.6. The vertical RMS for temperature (a) and salinity (b),where red line is the RMS of*
*misfits from AF, and blue line is the RMS of misfits from SF*
To show how the assimilation impacts the quality of temperature and salinity in the
North-west Pacific, the RMS differences between AF and SF for the 1-year




assimilation interval are displayed in Fig. 3.7 and 3.8. The statistics are divided into
two regions, the tropics (the south of 23.5°N, Fig. 3.7) and outside the tropics (the
north of 23.5°N, Fig. 3.8). The red line and blue line stand for the RMS of AF and SF
respectively in both figures ,.
In the tropics, the AF performs better and better than the SF over time. As shown in
the upper panel of Fig. 3.7, the RMS of AF and SF temperature misfits approximately
fit the observation equally well, only with the AF slightly closer to the observation
data. The RMS of AF is ~0.83 °C in 2006, which is improved by ~ 23.2% compared
to ~1.08 °C of SF. As shown in the lower panel of Fig. 3.7, the RMS of AF salinity
misfits performs better than the RMS of SF, with ~0.112 (PSU) of AF compared to
~0.138 (PSU) of SF.

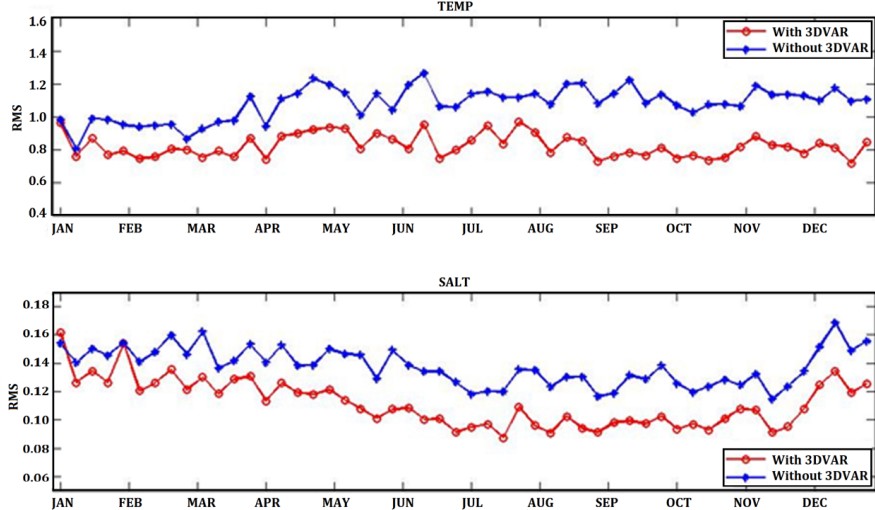


*Fig. 3.7.The RMS misfits for temperature ((a), in ℃) and salinity ((b), in psu) in tropic during*
*assimilation year (2006), where red line stands for RMS misfits with data assimilation and*
*blue line stands for RMS misfits without data assimilation.*
In the sub-tropic, the RMS of AF also performs a greater improvement than SF. As
shown in the upper panel of Fig. 3.8, the RMS of AF is ~1.43 °C in 2006, which is
improved by ~ 25.1% compared to ~1.91 °C of SF. As shown in the lower panel of
Fig. 3.8, the RMS of AF salinity misfits performs better than the RMS of SF, with
~0.135 (PSU) of AF compared to ~0.173 (PSU) of SF, which is improved by ~ 22.0%.


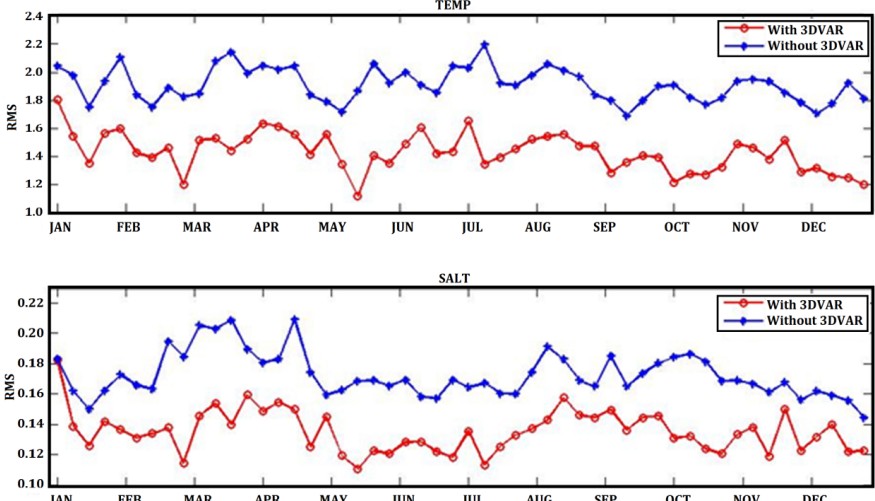


*Fig. 3.8. The RMS misfits for temperature ((a), in ℃) and salinity ((b), in psu) in sub-tropic*
*during assimilation year (2006), where red line stand for RMS misfits with data assimilation*
*and blue line stand for RMS misfits without data assimilation.*
**4. Summary**
In this paper, we implement 3DVAR on ROMS with the ability to assimilate the T&S
Argos profiles. The data assimilation system is tested on an eddy-resolving model of
the North-west Pacific. A specific feature of ROMS 3DVAR system is separating the
background error covariance matrix into vertical and horizontal modes in order to
reduce the order of the data assimilation. Horizontal covariance is modeled as
Gaussian function, whilst vertical covariance which is calculated from a long-term
model simulation is represented by Empirical Orthogonal Functions (EOFs).
The T&S of Argos profiles are assimilated into the North-west Pacific model for the
period of 2006. Results show that the assimilation system can get a beneficial effect in
the model region.
The analysis produced by the data assimilation has been validated by the monthly
means SST from satellite, which is an independent observation. In the model region,
the data assimilation system has the capability of "bringing" the model closer to the
observations.
Statistical indexes indicate that the RMS of misfits for temperature is less than 1.0 °C
in the tropics domain and less than 1.5 °C in the subtropics domain with the main





error from the Kuroshio Extension region. The RMS misfit salinity error is less than
0.15 PSU in the model region.
**Acknowledgments**
*This work was supported byThe National Natural Science Foundation of China under contract No.*
*41222038, 41206023 ; the National Basic Research Program of China ("973" Program) under*
*contract No. 2011CB403606; and the "Strategic Priority Research Program" of the Chinese Academy*
*of Sciences Grant No. XDA1102010403.*

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
