# Peer review of "Nat. Hazards Earth Syst. Sci. Discuss., doi:10.5194/nhess-2016-53, 2016 Manuscript under review for journal Nat. Hazards Earth Syst. Sci. Published: 10 March 2016"

_Natural Hazards and Earth System Sciences, 2016_

## Referee Comment (RC1) · Anonymous Referee #1 · 22 Apr 2016

The manuscript "Data assimilation of Argos profiles in North-west Pacific Model" by Wang et al uses a 3DVAR with a novel specification of the background error covariance to assimilate Argos profiles in the North-west Pacific and validates the performance. Overall the manuscript is not well-written, with many grammatical errors, the figure caption confusion and awkward wording throughout the text. It is my recommendation that this manuscript should be significantly revised before it is considered for publication.

**Major comments**

The background error covariance matrix B is one of important factors which affect the performance of the assimilation. A good B should reflect the flow-dependence of the background error covariance. In this manuscript, the authors mentioned a novel method used to estimate the B. More details should be given. E.g. How to calculate the VEOFs ? The VEOFS are associated with the univariate or multivariate? What is the structure of the B? What is the difference from the conventional B? What are the advantages of the new B in describing the structure of the background error covariance? Some figures are necessary.

To assess the performance of an assimilation method, the comprisons with the independent observations are necessary. The comparisons with the observations used in the assimilation only confirm the assimilation code, and make no significant sense. The independent observations are obtained by withdrawing some observations from the assimilation experiment. In the manuscript, the EN4 and MGDSST are used for comparisons. But, these are not independent observations (EN4 include Argo, MGDSST is the realxation data of the model). The comparisons to the independent observations should be added. The authors may validate the performance via many ways. E.g. The assimilation system is aimed for the operational forecasting. The authors may carry out some forecast or hindcast experiments with the initial conditions from the assimilation or no assimilation, and evaluate the forecast skill. The sense is very significant. Or the improvement on some physical phenomena or mechnisms are also convincing evidences.

Page 6, Section 3.1, the description throughout the paragraph 2 is confused. E.g.

"The model SST (Fig. 3.1(b)) is consistent with that derived from MGDSST (Fig. 3.1(a))." Firstly, in Bohai sea and Japan sea, the diffence reaches 2.5$^o$, is it consistent? Should point out the regions where the model is constistent with observations. Secondly, according to this statement, Fig. 3.1(a) should be MGDSST, but the figure caption indicates AF.   In the statement "In subtropical basins, temperature is generally high near the western boundary. While in sub-polar basins, the zonal temperature gradient reverses sign, with low temperature in the western basin.", accordint to the definition of the subtropic( north of 23.5N) in the manuscript, fig3.1 does not show high temperature near the western boundary in the subtropic basins, and the zonal temperature gradient reversed sign with low temperature in the western basin in the sub-polar basins. In the statement "Simulated SST is higher in   summer and lower in winter, compared to MGDSST (Fig. 3.1(d)).", where is figure 3.1(d)? Moreover, according to figure 3.1(c), the simulated SST is lower than MGDSST in summer(July) and higher in winter(January).

**Minor comments.**

1. Firstly, what is Argos? Is it the same as Argo? If yes, "Argo", instead of "Argos", should be used conventionally. If it is an abbreviation, please give full name.

2. How long is the assimilation window,   the frequency of the assimilaiton?

3. MGDSST should not be considered as independent observations, because it is the relaxation data. Many SST observations are available. Why to choose it?

4. Figure 3.1 should be SST, but the caption is at 150m. The depth 150m should not be as the sea surface. Please confirm it.

5. Why the SST after assimilation is higher than MGDSST while the SST without assimlation is lower in the Bohai Sea and Japan sea in July? Why is it not the case in other seasons?

6. The experiment without assimilaiton should be added to fig3.2

7. It is difficult to identify the difference between AF and EN4 from Fig 3.3 and 3.4 due to such a deep depth. The maximum of the observation depth is about 2000m,

and the impact of the assimilation on the upper ocean is greatest. To highlight the difference,    the depth of figure 3.3-3.4 should be limited to 1000m.

8. The figure captions are not clear. E.g. Fig3.5-3.6 should give the specific date. Fig 3.7-3.8 are related to the whole column or sst alone?

---

## Referee Comment (RC2) · Anonymous Referee #2 · 13 May 2016

General comments:

The aim of the paper is to show the development of an ocean operational system providing 5 days forecasts for the North-west Pacific using ROMS. Supposedly, a new method to determine the error-covariances for the Argo data is proposed, although it is not clearly shown in the text. The paper is poorly written, with grammatical errors that compromise its understanding and several points that need a deeper analysis and / or explanation. There is a lack of logic in the explanation of the experiments and the assimilation process, what makes it difficult to understand the reasoning for the methodology.

The assimilation process is poorly described. The "novel" background error covariance matrix determination method is not explained and the assimilated observations not

presented.

The validation of the model results is not properly done. Other quantitative tools, such as Taylor diagrams are more appropriate for the task. Adequate datasets must be used for validation, instead of optimally interpolated maps. Monthly mean properties at a set level and one section do not provide a robust validation of the model results. Moreover the model is in part "validated" against the assimilated observations, which is not a valid approach. One should use independent observations and/or the analysis of important oceanographic process to perform a proper validation.

Therefore, I can not recommend this manuscript for publication in its present form.

Detailed comments:

Line 44: It is simplistic to say the forecast quality depends on the initial conditions. The surface forcing and boundary conditions are also determinant and usually updated in the assimilation process. Therefore, an assimilation system that focus only on the determination of optimal initial conditions is limited and may not perform well.

Line 57: You should provide some information that justifies your statement that the 4DVar scheme is to expensive for an operational system. It is in fact broadly used in operational forecast systems for diverse regions (eg: Powell et al. 2008 – Ocean Modelling). A comparison between the computational cost of the 4DVar and the proposed 3DVar schemes is necessary.

Line 75: What do you mean by "multivariate properties"?

Line 89: How did you smooth your bathymetry? Is the total volume of the basin conserved? How different is the smoothed bathymetry against the reality? Please keep in mind that strong changes in the slopes and modifications of the basin volume can have important effects on the dynamics.

Line 91: You say velocities (I assume both barotropic and baroclinic) and water levels are provided by SODA in the open boundaries. What about temperature and salinity?

Line 106: In the "Assimilation algorithms" section you should explain how the error-covariance and EOFs are calculated. Moreover, the use of EOFs for obtaining the vertical structure is one of many possible approaches and its use should not be generalized for 3DVar schemes (line 124). This whole section consists in a poor explanation of the 3DVar approach and do not touch the main point that is how you use it to assimilate Argo. It lack important information, such as: A description of your dataset; definition of your error-covariance matrix; it is not clear if you are using strong or week constrain;

Line 154: If your focus is on 5-days forecasts, how can the comparison of monthly mean SSTs be significant? Moreover, 150m model temperatures are not representative of SST. Several datasets of SST, SLA, SSS and surface velocities are available for download for the validation of model results.

Line 175: At least difference maps for the AF and SF schemes as in Fig.3.1 should be generated to show some improvement of the assimilation results. Moreover, the use of optimal interpolation products is not recommended for the validation since it contains intrinsic decorrelation scales used in the interpolation. The validation should be done against real data!

Line 178: What does "very similar" means? You should avoid relative terms and show the actual numbers. A simple root-mean-squared deviation can give an idea of how close the model is to the observations.

Line 180: Why did you choose this section? You must justify it (Some process of particular interest?). The section must be shown in Fig. 2.1

Line 183-184: This phrase makes no sense...

Fig 3.3 and 3.4: Again, both AF and SF should be shown, together with difference plots.

Line 199 and Fig. 3.5: I assume by observations you mean Argo profiles. It is not a

valid validation the comparison with the observations that were assimilated. It does not show any improvement of your model solution. In addition, it should be clearly stated is these are mean profiles. If so, errorbars (standard deviation) must be added to the plots.

Section 3.2: This whole section is not valid, since independent observations must be used in the validation process.

---

## Author Comment (AC1) · 22 Jun 2016

Thanks for reviewers' comments of the manuscript. We will pay more attention to the writing approaches of the revised context. Major comments: The background error covariance matrix B is very important for the performance of the assimilation system. We are sorry for not providing a detailed description of B in the manuscript. In the data assimilation system, the background-error covariance matrix is decomposed into horizontal correlations and vertical covariances. Horizontal correlations are modeled using four iterations of a first-order recursive. The Empirical Orthogonal Functions (EOFs) is used to represent the vertical component of the background-error covariance matrix. The EOFs are calculated from daily simulations, which contain multivariates of sea surface height, temperature and salinity from 1995-2005 at full model resolution. There are 100 vertical levels for B. Some figures will be provide in the latest revised

manuscript. Main steps in the model are as follows: first compute the misfits between simulations and observations when the model runs; then run the data assimilation system while collect the increments matrix; at last, use the increments matrix to correct the initial condition of the model for next day's running. In the manuscript, EN4 and MGDSST are used for comparisons, since misfits are computed before the assimilation, observations can be independently used in the system. Therefore, Argo profiles and EN4 can be used as independent observations; while MGDSST cannot be used as independent observation because it has been relaxed into the model. So we consider replacing MGDSST with GHRSST. In order to validate the performance of the DA system, we will add a section to discuss the performances of forecast and hindcast using the initial conditions with or without assimilation. I'm sorry for the figure caption confusion. There is a great change in Bohai sea and Japan sea in July as shown in Figure 3.1, where the difference reachesto2.5°C. We will point out the regions where the model is consistent with observations. For the caption of Fig 3.1(a), it is Analysis Fields of SST, not the MGDSST. For the caption of Fig 3.1, we will pay more attentions to the description of it. Thanks for your detailed comments again. Minor comments: 1. We have used "Argo" instead of "Argos". 2. The assimilation window is daily. 3. We will use GHRSST instead of MGDSST for comparison. 4. The caption has been changed to surface. 5. We are not clear about the reason why. We will make several experiments to discuss about that. 6. The experiment without assimilation has been added to Fig. 3.2. 7. We will limit the depth of Fig.3.3-3.4 to 1000m. 8. We will re-plot the Fig. 3.5-3.6 and add the specific date. For the Fig. 3.7-3.8, they are related to whole column. A clearer description will be given in the revised manuscript. That's all. Thanks very much for the reviewer's comments again.

---

## Author Comment (AC2) · 22 Jun 2016

General comments:

Thanks for reviewers' comments of our manuscript. The background error covariances are not clearly shown in the manuscript. We will give a detailed introduction of it in the new manuscript. Some necessary figures will also be provided. Moreover, we will pay more attention on the writing approaches to fulfill the requirements of publishing. The assimilation process will be provided in the new manuscript. We use almost one year observation of Argo profiles in our experiment, but the assimilated observations are not presented in the manuscript, which will be added. Some other quantitative methods will be used to validate the performance of model. Because the Argo hasn't been assimilated in the model when computing the misfits between simulation and observation, we

believe that the observation of Argo profiles can be used independently.

Detail comments:

Line 44: The quality of initial conditions has a great influence on simulation results. A good initial conditions can improve the accuracy of numerical simulation. We know the surface forcing and boundary condition are also important. So the CFSR and SODA datasets, which are widely used reanalysis datasets in the world, have been used in the model.

Line 57: We notice that some operational forecast tests about 4dvar has been applied, but 3dvar can save more computation amount, and is more suitable for operational forecasting than 4dvar. The operational data assimilation system based on 4dvar scheme maybe our next work.

Line 75: "multivariate properties" here means validation of temperature and salinity.

Line 89: ROMS uses terrain-following coordinates. In order to reduce the influence of the seamount on the model stability, the bathymetry should be smoothed. In our model, there are used three parameters to smooth the bathymetry: the slope parameter (r=grad(h)/h) maximum value for topography smoothing, the number of pass of a selective filter to reduce the isolated seamounts on the deep ocean, and the number of pass of a single hanning filter at the end of the smoothing procedure to ensure that there is no 2DX noise in the topography. We selected the most effective parameters, which are not only maintain stability of model, but also close to the real topography to the greatest extend.

Line 91: The temperature and salinity are also provided by SODA.

Line 106: The formulation of the background term of the cost function has been described in Dobricic and Pinardi (2008). The background-error covariance matrix is decomposed into horizontal correlations and vertical covariances, which are assumed to be independent of each other. The horizontal correlations are supposed to satisfy

Interactive
comment

[Figure]

Gauss distribution with a constant correlation radius which is given as a parameter. The repeated application of Laplacian operator can model isotropic Gaussian spatial correlations. A first-order recursive filter with horizontally homogeneous is applied to the Laplacian operator for calculating the horizontal covariances. For the vertical component of the background-error covariance matrix, the Empirical Orthogonal Functions (EOFs) is used to represent. The EOFs are calculated from the model daily simulations, which contain three variables of sea surface height, temperature and salinity from 1995-2005 at full model resolution. Meanwhile, the strong constrain has been used in the model.

Line 154: We will consider to use more datasets for the validation of the model results.

Line 175: The difference maps for AF and SF will be plotted in Fig.3.1. At the same time, we will collect some new in-situ observations for validation.

Line 178: The word "very similar" isn't suitable in this paper. We will provide the actual values to show how close the model to the observations.

Line 180: The Japan Meteorological Agency has long time observations for the section of 137°E, but we can't download the observations in 2006. We believe the section of 137°E would be more suitable here, So the section of 137°E will be used to instead of section of 137°E. Furthermore, the location of the 137°E section will be shown in the Fig. 2.1.

Line 183-184: The phrase shall be deleted if no sense.

Fig3.3 and 3.4: Both figures will be re-plot to show the difference between AF and SF.

Line 199 and Fig 3.5: The Argo profiles haven't been assimilated into the model when comparison. We believe that the Argo profiles can be valid validation in the experiment. The mean profiles and standard deviations will be stated in the new manuscript.

Section 3.2: Because the Argo profiles are independent observations, the section 3.2 maybe is valid.

---

## Author Comment (AC4) · 10 Aug 2016

Thanks for reviewers' comments of the manuscript. We have paid more attention to the writing approaches of the revised context. Major comments: The specification of the background-error covariance matrix is one of the most important aspects affecting the performance of the assimilation system. The detailed description of B have been given in the revised manuscript. In the data assimilation system, the background-error covariance matrix is decomposed into horizontal correlations and vertical covariances. Horizontal correlations are modeled using four iterations of a first-order recursive. The Empirical Orthogonal Functions (EOFs) is used to represent the vertical component of the background-error covariance matrix. The EOFs were calculated from the daily means of a full-resolution model simulation covering 1995-2005, and contain covariances of sea level, temperature and salinity. Each monthly set consists of 20 EOFs

with 100 z-levels in the vertical. Fig. 3 shows the map of yearly mean background-error standard deviation reconstructed from the EOFs, where the Fig. 3(a) refers to sea level error, Fig. 3(b) to temperature at surface and Fig. 3(c) to salinity at surface. There are three main steps in the system as shown in Fig. 2: a) preparationof temperature and salinity observations from Argo profiles; b) integration of the NwPM model using the prevision day analysis increments to correct the initial condition; c) running the data assimilation system using the daily observation innovations (or misfits) to produce the new analysis increments for the next model integration. In this assimilation scheme, the Argo observations are not assimilated in the model when the misfits are computed. Therefore, the observation misfits are independent from the assimilation and can be used to validate the performance of the system, if temporal correlations of the observation errors are neglected as usual. In the section of model validation, the independent in situ observations, such as profiles, satellite data and reprocessed datasets, are used to validate the performance of data assimilation system. I'm sorry for the figure caption confusion, which has been corrected in the new manuscript. Thanks for your detailed comments again. Minor comments: 1. We have used "Argo" instead of "Argos". 2. The assimilation window is daily. 3. We have used OISST to instead of MGDSST for comparison. 4. The new Fig. 7 is used to instead of Fig. 3.1, and the caption has been confirmed. 5. In the Fig. 7, the depths below 200m has been masked out, because the Argo profiles mainly distributed in the region deeper than 2000 m. The Bohai Sea has been masked out in the new figure. In the experiment of 2010, the reductions of SST bias are similar in different seasons. 6. Fig. 10. monthly mean salinity bias at surface, is used to instead of Fig. 3.2, where the SSS error calculated by BF and EN4.0.2, and by AF and EN4.0.4 has been shown. 7. The depth of Fig. 11 and 12, which are used to instead of Fig. 3.3 and 3.4, has been limited to 1000 m. 8. Fig. 3.5 has been deleted in the manuscript, where has added the new Fig. 4 to show the vertical distribution of misfits for temperature and salinity. Fig. 6, which is used to instead of Fig. 3.7-3.8, are related to whole column. That's all. Thanks very much for the reviewer's comments again.